# Discriminant Analysis of Pu-Erh Tea of Different Raw Materials Based on Phytochemicals Using Chemometrics

**DOI:** 10.3390/foods11050680

**Published:** 2022-02-25

**Authors:** Shao-Rong Zhang, Yu Shi, Jie-Lin Jiang, Li-Yong Luo, Liang Zeng

**Affiliations:** 1College of Food Science, Southwest University, No. 2 Tiansheng Road, Beibei District, Chongqing 400715, China; qianzhihe225093@126.com (S.-R.Z.); shi3796@icloud.com (Y.S.); liyongluo1979@126.com (L.-Y.L.); 2Taetea Group, Menghai Tea Factory, Xishuangbanna Dai Autonomous Prefecture, Menghai 665000, China; dyjiangjielin@126.com; 3Tea Research Institute, Southwest University, No. 2 Tiansheng Road, Beibei District, Chongqing 400715, China

**Keywords:** Pu-erh tea, raw material, phytochemical, chemometrics analyses, discriminant model, generalization capability

## Abstract

Pu-erh tea processed from the sun-dried green tea leaves can be divided into ancient tea (AT) and terrace tea (TT) according to the source of raw material. However, their similar appearance makes AT present low market identification, resulting in a disruption in the tea market rules of fair trade. Therefore, this study analyzed the classification by principal component analysis/hierarchical clustering analysis and conducted the discriminant model through stepwise Fisher discriminant analysis and decision tree analysis based on the contents of water extract, phenolic components, alkaloid, and amino acids, aiming to investigate whether phytochemicals coupled with chemometric analyses distinguish AT and TT. Results showed that there were good separations between AT and TT, which was caused by 16 components with significant (*p* < 0.05) differences. The discriminant model of AT and TT was established based on six discriminant variables including water extract, (+)-catechin, (−)-epicatechin, (−)-epigallocatechin, theacrine, and theanine. Among them, water extract comprised multiple soluble solids, representing the thickness of tea infusion. The model had good generalization capability with 100% of performance indexes according to scores of the training set and model set. In conclusion, phytochemicals coupled with chemometrics analyses are a good approach for the identification of different raw materials.

## 1. Introduction

Pu-erh tea is defined as a geographical indication product by the General Administration of Quality Supervision, Inspection, and Quarantine of the People’s Republic of China (GB/T 22111-2008), and is one of the most popular tea beverages in Asian countries; in particular, southwestern China and South Asian countries attribute this to its unique flavors and beneficial effects on human health [1]. It is processed from the sun-dried green tea leaves and can be classified into ancient tea (Gu-shu cha, AT) and terrace tea (Tai-di cha, TT) based upon the source of raw material [2,3]. AT is collected from ancient tea gardens that have good economic and ecological efficiencies such as climate regulation, water and soil conservation whereas TT is gathered from terrace tea plantations that rely on the good management of tea fields including fertilization, pruning, and pesticide spraying. The differences in growth environments and management methods of both result in different flavors: compared with TT, AT has a richer taste with durability and a more distinctive aroma, which is widely considered to have more preservation value [4]. However, their similar appearance has led to low market identification of AT, which seriously damages the interests of consumers and the reputation of tea producers. Meanwhile, lack of yield and the unclear relevant laws or market regulation of AT have also created an opportunity for tea adulteration [5].

Based on the phenomenon, numerous researchers have investigated the differences of AT and TT from phytochemicals, soil nutrients, and flavor, but most studies have simply compared them based on a single component because there is still a lack of methods to systematically identify the differences between AT and TT in the case of multiple samples and multiple indicators [6,7,8,9]. Lin et al. [10] established discriminant models of AT and TT through rare earth elements, but these components were inapplicable and difficult to understand for tea drinkers, especially consumers without professional knowledge in the tea market. Therefore, it is necessary to develop a valid method through more intuitive indicators to scientifically distinguish AT from TT. Chemometric analyses are a data analysis tool to extract effective information from multivariate chemical data. This is often combined with other methods such as high-performance liquid chromatography (HPLC) and mass spectrometry to analyze food adulteration in the market and explore whether there are similar characteristics of unknown samples [11,12]. Principal component analysis (PCA), hierarchical clustering analysis (HCA), and the establishment of discriminant model are the main chemometric techniques [13]. According to the results on discriminating Brazilian propolis using chemometrics by AF Mottese [14], it was found that DA/DTA was complementary to PCA/HCA and the important information of the discriminant model could be obtained based on them. At present, they have been widely applied in the identification of food adulteration in products such as butter, camellia oil, honey, and tea [15,16,17,18,19,20,21]. For example, Zhou et al. [18] correctly identified Pu-erh ripened teas of different production origins based on PCA and HCA; Aboulwafa et al. [22] studied the quality of green tea samples from the South and the East Asian regions, and the results showed that there were good separations of samples between them by establishing the model. Phytochemicals including phenolics, free amino acids, caffeine, and other components could intuitively reflect tea quality through flavor [23] whereas their combination could prove to be a useful tool to distinguish Pu-erh tea made from different raw materials.

Hence, with regard to the scientific classification of Pu-erh tea from different raw materials and guide tea producers and consumers to effectively identify tea products, 30 ATs and 50 TTs from Yunnan Province were collected in this work, and 26 phytochemicals including three mixtures, six catechin components, two purine alkaloids, and 15 free amino acids were determined by spectrophotometry method and high-performance liquid chromatography (HPLC), respectively. Further chemometric analyses were used to reveal classification accuracy and construct a model.

## 2. Materials and Methods

### 2.1. Chemicals

Used chemicals were of analytical grade unless otherwise stated. Folin–Ciocalteu reagent, sodium carbonate (NaCO_3_), methanol, ninhydrin, stannous chloride (SnCl_2_), disodium phosphate (Na_2_HPO_4_), potassium dihydrogen phosphate (KH_2_PO_4_), 5-sulfosalicylic acid, and glacial acetic acid were purchased from Taixin Chemical Company (Chongqing, China), among which methanol and glacial acetic acid were HPLC-grade. Gallic acid and glutamate were obtained from the Kelong Chemical Factory (Chengdu, China). Aspartic acid (Asp), serine (Ser), glutamic acid (Glu), glycine (Gly), alanine (Ala), cysteine (Cys), valine (Val), isoleucine (Ile), leucine (Leu), tyrosine (Tyr), phenylalanine (Phe), lysine (Lys), histidine (His), arginine (Arg), and theanine were guaranteed reagents purchased from Seebio Technology Co. Ltd. (Shanghai, China). Caffeine, (−)-epigallocatechin gallate (EGCG), (−)-gallocatechin gallate (GCG), (−)-epicatechin gallate (ECG), (−)-epigallocatechin (EGC), (−)-epicatechin (EC), and (+)-catechin (C) were guaranteed reagents purchased from Chengdu Biopurify Phytochemicals Ltd. (Chengdu, China). Theacrine was obtained from Better-in Com (Shanghai, China).

### 2.2. Tea Samples

As shown in Table 1, a total of 80 samples including 30 ATs and 50 TTs were collected from five ancient tea gardens (including Banpen, Laobanzhang, Hekai, Xinbanzhang, Laoman’e) and nine terrace tea plantations (including Nannuo, Bulang, Mensong, Lincang, Lancang, Xiding, Gelanghe, Menla, Dali) in Yunnan Province. The raw Pu-erh samples were processed through five steps including picking, withering, green removing, rolling and twisting, and sun-drying, and marked with the corresponding numbers according to the sample names before grinding. All samples were stored at −4 °C until further analysis.

### 2.3. Determination of Phytochemicals

#### 2.3.1. Water Extract (WE)

WE in the AT and TT was detected by the constant temperature drying method [24] with some modifications. Briefly, 1 g of the ground sample (m_0_) was soaked in 150 mL boiled distilled water for 45 min. After washing and filtration using 75 mL boiled distilled water, tea grounds were put in an oven (120 ± 2 °C) to bake for 4 h before weighing (m_1_). The WE content was expressed as (m_0_ − m_1_) × 1000/m_0_ mg/g.

#### 2.3.2. Total Phenolics (TPC)

The determination of TPC content was performed using Folin–Ciocalteu reagent [25]. Briefly, 0.2 g of the ground sample and 5 mL of 70% methanol (70 °C preheat) were placed in a 70 °C water bath pot for 10 min and centrifuged at 3500 rpm for 10 min after being cooled. The above operation was repeated, and all supernatants were merged to a constant volume of 10 mL. Then, 1 mL of the sample or water, appropriately diluted, was taken, followed by 5 mL of 10% Folin–Ciocalteu reagent. After 5–8 min at room temperature, 4 mL of 7.5% sodium carbonate solution was added and the mixture was placed at 25 ± 2 °C in the dark for 1 h. The absorbance was measured at 765 nm using a Synergy H1MG microplate reader (Synergy H1MG; BioTek Instruments Inc., Winooski, VT, USA). Gallic acid (0–0.0055 mg/g) was used as the reference standard, and the results were expressed as gallic acid equivalents per gram sample (mg/g).

#### 2.3.3. Total Free Amino Acids (TFAAs)

The TFAAs contents of samples were identified using the ninhydrin colorimetric assay [26]. Briefly, 1 g of the ground sample and 150 mL water were placed in a boiling water bath for 45 min, and filtrated through decom-pressure filtration was added to water, yielding a 150 mL volume. Then, a 1 mL of sample was taken, followed by 0.5 mL phosphate buffer (pH 8) and 0.5 mL 2% ninhydrin. After shaking, the solution was placed in a boiling water bath for 15 min. Then samples were cooled to 25 ± 2 °C and water was added to a 25-mL volume before measurement at 570 nm using a Synergy H1MG microplate reader (Synergy H1MG; BioTek Instruments Inc., Winooski, VT, USA). Glu (0–0.6 mg/g) was used as the reference standard, and the results were expressed as Glu equivalents per gram sample (mg/g).

#### 2.3.4. Catechins Components and Caffeine

Catechin components (EGCG, GCG, ECG, EC, EGC, C) and caffeine were identified using the high-performance liquid chromatography (HPLC) method [27]. Briefly, 0.2 g of the ground sample and 5 mL of 70% methanol (70 °C preheat) were placed in a 70 °C water bath pot for 10 min, and centrifuged at 3500 rpm for 10 min after being cooled. The above operation was repeated, and all supernatants were merged to a constant volume of 10 mL. The above solution was filtered through a 0.45-µm organic membrane (Jinteng Experimental Equipment Co. Ltd., Tianjin, China) until further HPLC analysis. The chromatographic conditions were as follows: Agilent ZORBAX SB-C18 (5 μm, 4.6 mm × 250 mm); detection wavelength at 278 nm; mobile phase A, 0.2% glacial acetic acid in water; mobile phase B, methanol; flow rate, 0.9 mL/min; column temperature, 35 °C; and injection volume, 10 μL; gradient elution, 18–25% B, 0–25 min; 25–35% B, 25–30 min; 35–18% B, 30–32 min; and 18% B, 32–37 min. Catechin components and caffeine levels were expressed as mg/g.

#### 2.3.5. Theacrine

The theacrine in the AT and TT was identified according to Li et al. [28] with some modifications. Briefly, the ground sample was extracted with 75% alcohol in a ratio of 1:20 (*w*/*v*) for 10 min and centrifuged at 12,000 rpm for 10 min. Then, the obtained supernatant was filtered through a 0.45-µm organic membrane (Jinteng Experimental Equipment Co. Ltd., Tianjin, China) until further HPLC analysis. The chromatographic conditions were as follows: Agilent ZORBAX SB-C18 (5 μm, 4.6 mm × 250 mm); detection wavelength at 280 nm; mobile phase A, acetonitrile/acetic acid/water (3:0.5:96.5, *v*/*v*/*v*); mobile phase B, acetonitrile/acetic acid/water (30:0.5:69.5, *v*/*v*/*v*); flow rate, 1.0 mL/min; column temperature, 32 °C; and injection volume, 10 μL; gradient elution, 20–80% B, 0–35 min; 20% B, 35.01 min; 20% B, 35.01–40 min. The theacrine level was expressed as mg/g.

#### 2.3.6. Free Amino Acids

The free amino acids were detected according to Lu et al. [29] using an Amino Acid Analyzer (L-8900, Hitachi, Tokyo, Japan). Briefly, a 4 mL sample and 4 mL 10% sulfosalicylic acid were added to a 10-mL tube. After one night, the sample was filtered through a 0.45-μm organic membrane (Jinteng Experimental Equipment Co. Ltd., Tianjin, China) before analysis. The Amino Acid Analyzer system used a mobile phase involving lithium citrate and UV–Vis detection at 440 nm and 570 nm. The flow rates were 0.35 mL/min for the mobile phase and 0.3 mL/min for the derivatization reagent. The column temperature was set to 50 °C, and the post-column reaction equipment was maintained at 135 °C. The temperature of the autosampler was kept at 4 °C, and the injection volume was 20 μL for the standard and samples. The free amino acids levels were expressed as mg/g.

### 2.4. Chemometric Analyses

All data were subjected separately to PCA and HCA to detect whether phytochemicals could be used to classify AT and TT. HCA and PCA were respectively performed using IBM SPSS Statistics software (Version 22, SPSS Inc., Chicago, IL, USA). An analysis of variance (ANOVA) was also used by IBM SPSS Statistics software (Version 22, SPSS Inc., Chicago, IL, USA) to determine significant differences (*p* < 0.05). Regarding the establishment of discriminant models, stepwise Fisher discriminant analysis (SFDA) was performed based on the model set. The model set consisted of 20 ATs and 40 TTs chosen randomly from 80 samples, and the other samples (10 ATs and 10 TTs) were the training set to check the accuracy of the model by the leave-one-out method (LOO). Based on discriminant variables from SFDA, DTA was further used to detect the classification of AT and TT, thereby establishing the model. The optimal model of AT and TT was selected by performance indexes and distance measurement. Both SFDA and DTA were applied by IBM SPSS Statistics software (Version 22, SPSS Inc., Chicago, IL, USA), and the evaluation performance indexes including accuracy, precision, recall, and F-score were calculated through the confusion matrix, as shown in Table 2. The formulas were as follows [30]:Accuracy = (TP + TN)/(TP + FN + FP + FN)(1)
Precision = TP/(TP + FP)(2)
Recall = TP/(TP + FN)(3)
F-score = 2TP/(2TP + FP + FN)(4)

## 3. Results and Discussion

### 3.1. Feasibility of Phytochemicals to Classification of AT and TT

Based on WE, TPC, TFAAs, six catechin components, two purine alkaloids, and fifteen free amino acids, PCA was used to explore whether the AT and TT could be distinguished through phytochemicals in this study. As shown in Table 3, the seven principal components (PCs) had an eigenvalue greater than 1 and explained 80.352% of the cumulative percentage of variance, among which PC1, PC2, and PC3 extracted according to the Kaiser criterion represented 24.542%, 19.348%, and 11.261%, respectively, of the variability of raw materials. Obviously, the first three PCs were the main components [31]. Figure 1a shows the score plots of PC1 versus PC3 and it could be observed that all samples were distributed in three areas, among which 11 TTs (T6, T7, T14, T15, T16, T17, T22, T23, T32, T46, and T48) were distributed in an area exhibiting positive scores of PC1, and other samples with cross distribution in the remaining two areas showed negative scores of PC1. Further combined with 3D score plots of PC1 versus PC2 versus PC3 (Figure 1b), it was found that 30 ATs were distributed around with TTs as the center, among which the 12 ATs (A29, A30, A13, A24, A27, A22, A28, A26, A21, A23, A20, and A25) in the left-upper corner showed negative scores of PC2 and positive scores of PC3; five ATs (A17, A18, A15, A16, and A19) in the right-upper corner showed positive scores of PC2 and PC3; and 13 ATs (A14, A11, A9, A3, A10, A12, A1, A6, A4, A7, A2, A8, and A5) in the right-lower corner showed positive scores of PC2 and negative scores of PC3.

Meanwhile, HCA also showed that when the square Euclidean distance was set at 14, all samples could be divided into eight clusters (clusters-A, B, C, D, E, F, G, and H) (Figure 2). 30 ATs were distributed in clusters-A, D, and H, corresponding to the sample distribution in the blue, orange, and red circles in Figure 1b, respectively. However, A19 was distributed in the red circle (cluster-H) in Figure 1b in the PCA while it was classified into cluster-D in the HCA. The reason might be that the new variables generated in the dimensionality reduction process of PCA produced different results. The clusters-B, C, E, F, and G included seven, twenty-one, two, nine, and eleven TTs, respectively, among which clusters-F and G corresponded to the sample distribution in the gray and green circles in Figure 1a,b, respectively; while other samples in Figure 1b corresponded to clusters-B, C, and E. In short, PCA and HCA can be used to explain whether teas could be distinguished according to season, year, production place, processing technologies, and category variations [32,33,34,35]. In this study, the results revealed that phytochemicals were the dominating influence factors in the classification of AT and TT, which could be used as identification indicators of both.

### 3.2. Comparison of Phytochemicals Differences Related with a Classification of AT and TT

The phytochemical contents related to the classification of AT and TT are presented in Appendix A (Appendix A), and the differences were analyzed based on the average levels of both. As shown in Figure 3a, the average levels of WE and TFAAs in AT were significantly (*p* > 0.05) higher than those in TT, whereas there was no significant difference in the average level of TPC between AT and TT. In terms of catechin components (Figure 3b), the average level of EC in AT was up to 34.3 mg/g, which was significantly higher than that in TT, while the contents of EGC, C, and GCG in AT were significantly lower than that in TT. Theacrine, as a kind of purine alkaloid, had a similar chemical structure to caffeine. It is the key component of Yunnan Kucha (also called bitter Pu-erh tea or Pu-erh Kucha tea), which not only has a beneficial effect on the human body but also a more bitter taste than caffeine [36,37]. As presented in Figure 3b, the level of theacrine in AT was significantly higher than that of TT, which indicated that the bitterness of AT was more remarkable than that of TT. Compared with the content of theacrine, the level of caffeine of AT and TT reached 22.6 and 61.6 folds, respectively, among which the content of caffeine in AT was significantly higher than that in TT. Additionally, free amino acids of AT and TT were dominant with theanine and Glu, according to Figure 3c. Theanine in AT and TT was up to 12.0 mg/g and 7.96 mg/g, respectively, and there was a significant difference between both; Glu in AT and TT had no significant difference. Asp and Ser were detected only in TT. Those findings were similar to those of Zhang [38] but opposite to the results on the levels of WE and TFAAs in AT and TT as reported by Liang et al. [6]. The disagreement might be due to the influence of processing technology.

Overall, growth environments and management methods had definite impacts on phytochemicals, resulting in significant differences in WE, TFAAs, four catechin components, two purine alkaloids, and eight free amino acids between AT and TT. Previous studies have also indicated that chemical fertilizers and environment could affect the contents of polyphenols, WE, caffeine, and free amino acids [7,39,40,41]. This indicates that the above differential components of AT and TT were representative, which could be considered as major components of classifying AT and TT in this study.

### 3.3. Establishment and Optimization of Discriminant Model of AT and TT

#### 3.3.1. Establishing a Discriminant Model through SFDA

SFDA is a popular recognition method by dimensionality reduction based on variance analysis and is one of the most effective methods for feature extraction [42]. To scientifically classify AT and TT, this study extracted key identification indicators by establishing a discriminant model based on 16 phytochemicals. Twenty ATs and forty TTs selected from all samples formed a model set and then the feature components of the model set were extracted, thereby establishing a discriminant model. The results indicated that there were eight phytochemicals that significantly (*p* < 0.05) affected the discriminant effect, and the order of influence degree was as follows, according to F value: EC > C > theanine > WE > EGC > theacrine > Ala > Arg (Table 4). However, due to the lack of detection of Ala and Arg in some samples, which could affect the accuracy of identification between AT and TT, six phytochemicals were finally used to establish a discriminant model in this study. The model is shown in Equation (5): the unknown sample can be determined as AT if Y > 0; otherwise, it was judged to be TT.
Y = −21.685 + 0.038 × WE − 0.094 × C + 0.12 × EC − 0.042 × EGC + 0.371 × theacrine + 0.217 × theanine(5)

#### 3.3.2. Optimizing Classification Model through DTA

To reflect the classification of samples more intuitively, DTA was used to detect the classification of 80 samples based upon six variables of the discriminant model. As shown in Figure 4, the tree involved a three-level structure with a total of eight decision nodes and four classification rules created by only using three elements. Among these, the decision rules were based on two concentration ranges of EC, four concentration ranges of C, and two concentration ranges of WE, which correctly identified 30 ATs and 50 TTs. According to the DTA results, the discriminant model of AT and TT was optimized. As presented in Equation (6), the unknown sample could be determined as AT if Y > −1; on the contrary, it was judged to be TT.
Y = −21.685 + 0.038 × WE − 0.094 × C + 0.12 × EC(6)

#### 3.3.3. Evaluating Classification Models

Upon the above two models, this study tried to evaluate the model externally and internally through a training set (10 ATs and 10 TT, *n* = 20) and model set (20 ATs and 40 TTs, *n* = 60) to explore the generalization capability of two discriminant models and choose the optimal model. As shown in Table 5, there were no misjudgments in the training set and model set by model calculation and the leave-one-out method (LOO), regardless of Equation (1) or Equation (2), that is to say, the accuracy, precision, recall, and F-score reached 100%. This suggested that the two discriminant models had a good generalization capability [43]. However, the above performance indexes were unable to help us choose the most suitable discriminant model of AT and TT. The distance measurement could be used to evaluate the separation effect of a model by a minimum distance of two categories: The larger the distance, the easier the classification and a lower error rate [44]. According to Figure 5a,b, the separation effect of AT and TT based on Equation (1) was better than that based on Equation (2) (1.60 > 1.11). This indicates that Equation (1), established by six components, was optimal, which could be used as the discriminant model of AT and TT in this study.

Regarding the six discriminant variables, previous studies have shown that WE is made up of soluble substances such as phenolics and alkaloids, reflecting the thickness of tea infusion [45]. Theanine and nonester catechins (EC, C, and EGC) were related to umami/sweetness and bitterness/astringency, respectively, which were significantly affected by some factors such as geographical environment, light, cultivar, and fertilizer [46,47]. For instance, the climate had an impact on the chlorophyll contents, thereby regulating the contents of nonester catechins; the contents of catechins were higher in northern areas while the contents of free amino acids were higher in the southeast. Theacrine significantly affected the bitter taste of tea infusion and had many excellent pharmacological effects such as sedative and hypnotic [48,49,50]. At present, our laboratory has already published relevant studies on the recognition threshold and leaching rule of theacrine [51]. The results showed that the taste of AT was more bitter than that of TT due to its high level of theacrine at the same recognition threshold. Meanwhile, the slow leaching rate of theacrine contributed to the endurance property in brewing AT. Overall, the six components could well reflect the differences between AT and TT, and an in-depth study on flavor contribution will be conducted based on these in the future, thereby providing a reference for regulating the sale of AT.

## 4. Conclusions

All in all, according to the 26 phytochemicals determined by spectrophotometry methods and HPLC, the PCA and HCA could divide 80 samples into AT and TT, and ANOVA showed the growth environment and management method caused the significant (*p* < 0.05) differences of the 16 phytochemicals as the principal factor including WE, TFAAs, four catechin components, two purine alkaloids, and eight free amino acids. Based on the ANOVA results, the discriminant model of AT and TT was eventually established based on six components including WE, EC, C, EGC, theacrine, and theanine by comparing the separation effect of SFDA and DTA. The accuracy, precision, recall, and F-score of the model were up to 100%, which illustrates the good generalization capability of the discriminant model. This study offers data support for Pu-erh tea from different raw materials from the perspective of phytochemical components and an analytical thinking of classification, which achieved a great effect. In future production practice, the classification method could be applied to classify and distinguish unknown samples. In addition, the chemometric analyses will also be a powerful tool in food fraud such as tea origin, storage time, and organic tea.

## Figures and Tables

**Figure 1 foods-11-00680-f001:**
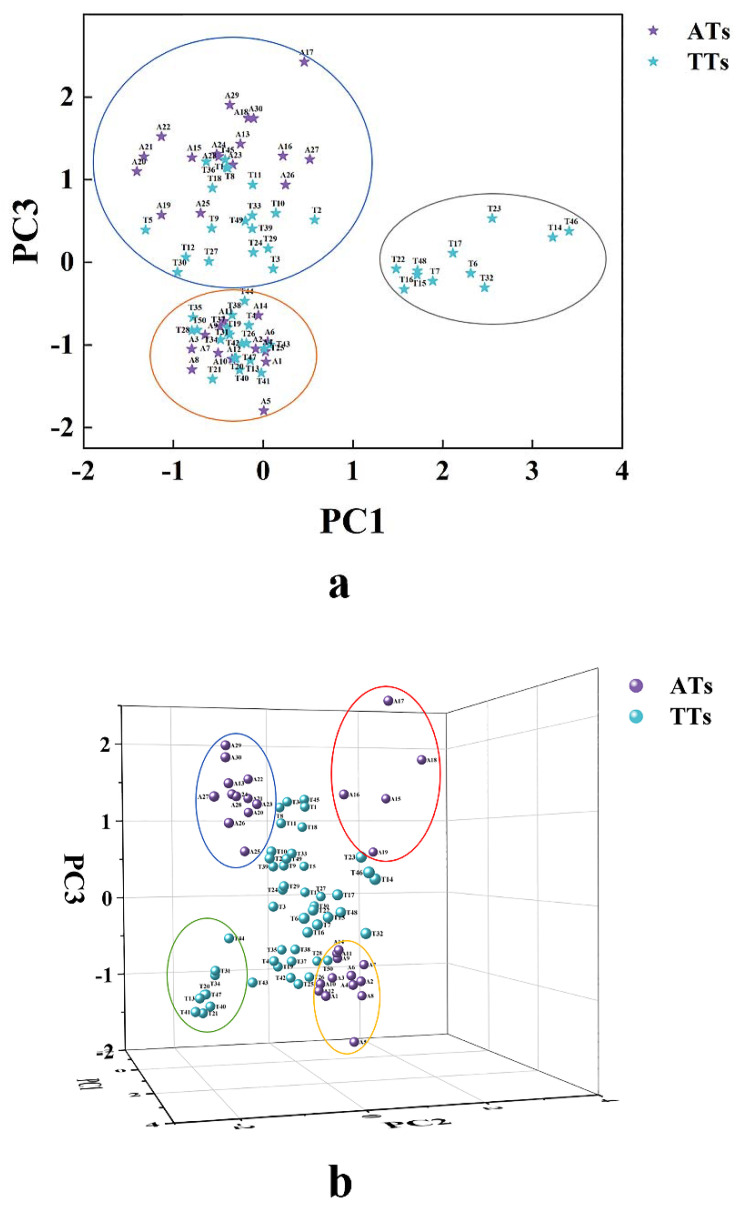
The score plots from the PCA of the phytochemicals in the ancient tea (AT, purple) and terrace tea (TT, cyan). (**a**) PC1 vs. PC3; (**b**) PC1 vs. PC2 vs. PC3. (**a**) had three clusters (blue, orange, and gray circle), among which the gray circle only contained TT. The blue, red, and orange circles only contained AT and the green circle only contained TT in (**b**).

**Figure 2 foods-11-00680-f002:**
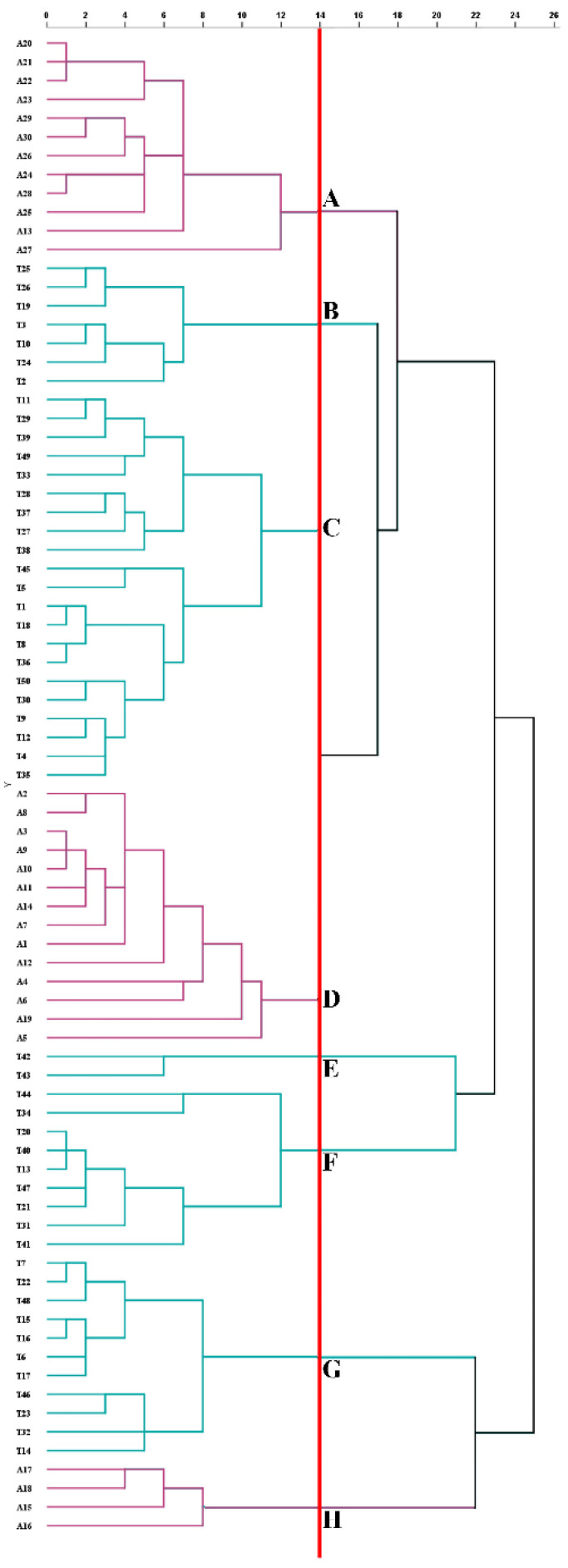
Cluster results on phytochemicals of ancient tea (AT, purple) and terrace tea (TT, cyan). The red line illustrates that samples were divided into eight clusters when the Euclidean distance squared was 14 including clusters-A, B, C, D, E, F, G, and H. Thirty ATs were distributed in clusters-A, D, and H and 50 TTs were distributed in clusters-B, C, E, F, G, and H.

**Figure 3 foods-11-00680-f003:**
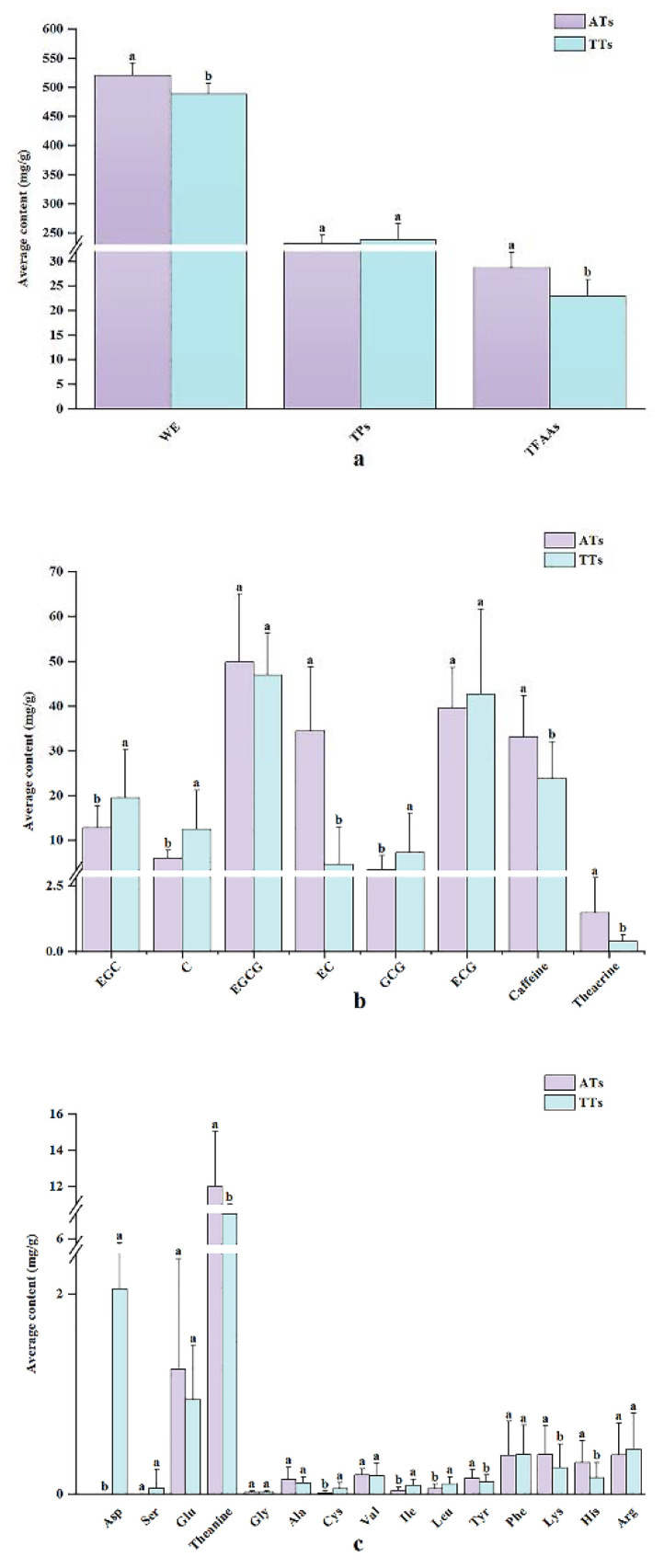
The average levels of the main phytochemical components of ancient tea (AT, purple) and terrace tea (TT, cyan) from Yunnan Province. (**a**) Mixtures; (**b**) catechin components and alkaloids; (**c**) free amino acids. The height of the error bar is ±standard deviation. Different letters with different superscripts on the same index represent significant differences (*p* < 0.05).

**Figure 4 foods-11-00680-f004:**
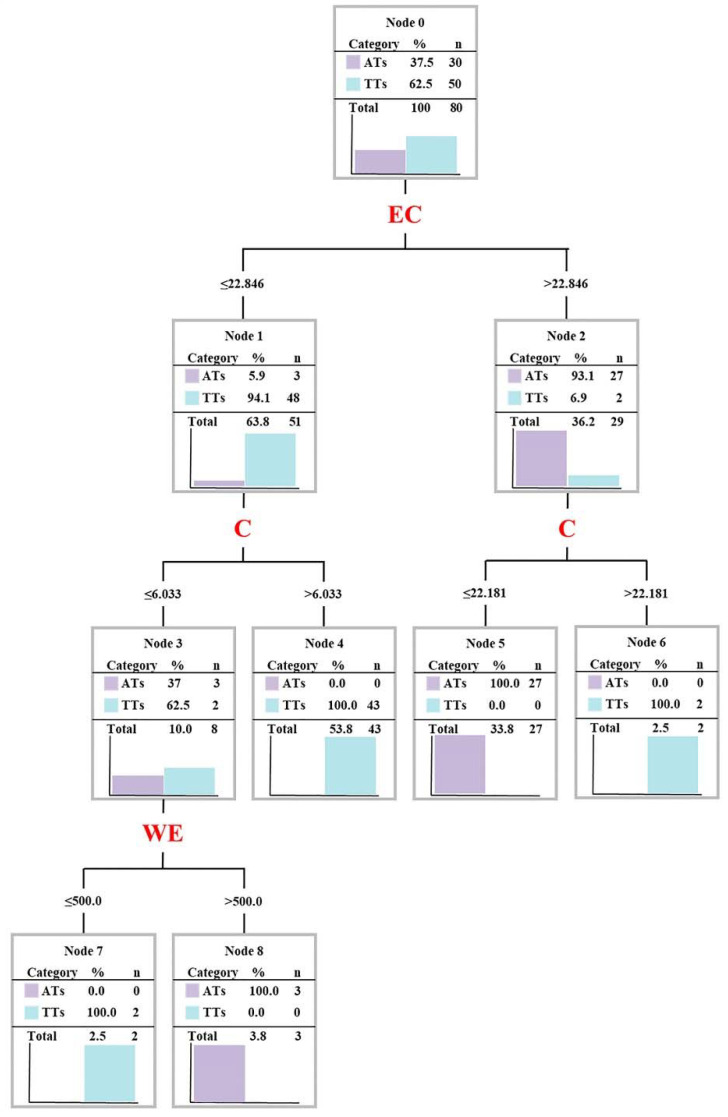
Decision classification tree resulting from the application of the DTA algorithm for the classification of ancient tea (AT) and terrace tea (TT) using the element profile. Concentrations are reported in mg/g.

**Figure 5 foods-11-00680-f005:**
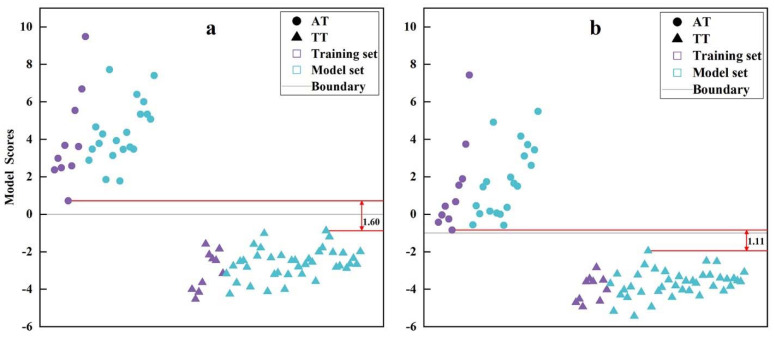
Model scores of the training set and model set calculated through classification model. (**a**) The classification model established through stepwise Fisher discriminant analysis (SFDA). (**b**) The classification model established through decision tree analysis (DTA). The circle represents AT and the triangle represents TT. The purple represents the training set and the cyan represents the model set. The gray line represents the dividing line of AT and TT. The interval between two red lines represents the minimum separation distance of AT and TT ((**a**), 1.60; (**b**), 1.11).

**Table 1 foods-11-00680-t001:** The sources, counts, and codes of different raw materials in Yunnan Province.

Raw Materials	Sources	Counts	Codes ^1^
AT	Banpen, Laobanzhang, Hekai, Xinbanzhang, Laoman’e	30	A1~A30
TT	Nannuo, Bulang, Mensong, Lincang, Lancang, Xiding, Gelanghe, Menla, Dali	50	T1~T50

^1^ The codes of AT and TT were respectively named as the uppercase of the first letter of them, followed by a number.

**Table 2 foods-11-00680-t002:** Confusion matrix table.

		True Classification ^1^	
		Type 1	Type 2
Predictive classification	Type 1	TP	TN
	Type 2	FP	FN

^1^ TP: true positives, represented the correct numbers of type 1 from the model prediction; TN: true negatives, represented the number of mistakes in type 1 from the model prediction; FP: false positives, represented the correct numbers of type 2 from the model prediction; FN: false negatives, represented the number of mistakes in type 2 from the model prediction.

**Table 3 foods-11-00680-t003:** Eigenvalue and variance contribution rates of PCA in AT and TT.

PCs	Eigenvalue	Percentage of Variance (%)	Cumulative Percentage of Variance (%)
PC1	6.381	24.542	24.542
PC2	5.030	19.348	43.889
PC3	2.928	11.261	55.151
PC4	2.534	9.745	64.896
PC5	1.745	6.711	71.607
PC6	1.205	4.635	76.242
PC7	1.069	4.110	80.352

**Table 4 foods-11-00680-t004:** Results and significance of variables affecting AT and TT discriminations by SFDA extraction.

Numbers	Variables	Statistics	df1	df2	Significance
1	EC	117.094	1	58	1.54 × 10^−15^
2	C	118.649	2	57	4.81 × 10^−21^
3	Theanine	106.949	3	56	3.67 × 10^−23^
4	WE	94.932	4	55	5.10 × 10^−24^
5	EGC	88.306	5	54	9.73 × 10^−25^
6	Theacrine	83.408	6	53	3.24 × 10^−25^
7	Ala	77.816	7	52	2.74 × 10^−25^
8	Arg	80.700	8	51	3.10 × 10^−26^

**Table 5 foods-11-00680-t005:** Confusion matrix obtained for the discriminant model AT and TT and evaluation of the model performance indexes.

Verification Modes			True Classification		Performance Index (%)			
			ATs	TTs	Accuracy	Precision	Recall	F-Score
Predictive classification	Training set	ATs	10	0	100	100	100	100
		TTs	0	10				
	Model set (LOO)	ATs	20	0	100	100	100	100
		TTs	0	40				
	DTA	ATs	30	0	100	100	100	100
		TTs	0	50				

## Data Availability

No new data were created or analyzed in this study. Data sharing is not applicable to this article.

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
