# Peer review of "Discriminant Analysis of Pu-Erh Tea of Different Raw Materials Based on Phytochemicals Using Chemometrics"

_foods, 2022, doi:10.3390/foods11050680_

Round 1

Reviewer 1 Report

I reviewed the manuscript entitled, Discriminant research for Pu-erh tea from different raw materials using chemometrics analyses based on phytochemicals. Methodology should be provided in detail. Results and discussion are appropriate with relevant literature.

Please change the title as follows: Discriminant analysis of Pu-erh tea of different raw materials based on phytochemicals using chemometrics

Line 123: Catechin components (EGCG, GCG, ECG, EC, EGC, C) and caffeine determination method should be described in detail

Line 56-63: this is very little introduction of chemometrics. Detailed info on PCA, HCA and discriminant model applications should be addressed with examples.  What is the need of chemometrics should be addressed in detail  

Since the study related to chemometric analysis, detailed methodology for PCA, HCA and discriminant model should be provided

Find the attached minor comments

Author Response

Thanks very much for taking the time to review this manuscript and I really appreciate all you comments and suggestions! Please find my itemized reponses in the attachment.

Reviewer 2 Report

The authors present a chemometric analysis based on phytochemicals to discriminate different types of teas.

The article is interesting and well planned.

There are, however, some issues that need to be addressed:

  1. First, the abstract is confusing, and the authors should eliminate the equation describing the discrimination model and only the six phytochemicals used in this equation. However, water extract is not a phytochemical, which should be clarified.
  2. Line 42: abundant taste is not clear. Did the authors mean richer flavour or tastier? Please clarify.
  3. Lines 53 and 54: This sentence is repetitive. Please rephrase it.
  4. Materials and methods. Please eliminate the extra dots on the sub-sections titles.
  5. Line 92: Were the tea samples ground as collected or previously dried before grinding?
  6. Was water extract detected? This is not clear. Does the water extract corresponds to total soluble solids? In this case, WE was measured or calculated. Please clarify.
  7. The abbreviation for total phenolics is usually TPC for Total Phenolic Content. Please change accordingly.
  8. The authors used 1 mL of sample to determine TPC and TFAAs. How was this sample prepared? From the Water Extract? Or from the tea leaves? 
  9. Line 130: The authors used the tea powder to analyse theacrine. This tea powder was the roasted tea leaves described in subsection 2.3.1 or dried tea leaves? Please clarify.
  10. The authors used Euclidean distance to divide samples into clusters. However, the authors should also use other distances such as Canberra or Manhattan to divide the cluster and assess which is the one more appropriate for this division.
  11. Line 245: Please correct the word "method".
  12. Line 305: please eliminate the journal name. Just indicate that this study was already published.
  13. Line 317: Please correct ANONA to ANOVA.
  14. Conclusions: The authors suggest that this method could help differentiate tea origins. Do the results obtain support this conclusion?

Author Response

(The authors gave the same response as above.)

Round 2

Reviewer 1 Report

Authors now answered the questions raised by me.

Author Response

Response to Reviewer 1 Comments

Comments and Suggestions for Authors:

Authors now answered the questions raised by me.

Thank you very much for taking the time to review our revised manuscript (foods-1601874). Your comments and suggestions are very helpful for us to improve our manuscript.

Reviewer 2 Report

The authors answered appropriately to the reviewer's comments. 

The article is improved and minor issues are still presented:

  1. Abstract line 28: Please eliminate the word and
  2. Line 91: please change place to origin.
  3. Line 1262 to 130: Please correct the sentence to " The raw Pu-erh samples were processed through five steps including picking, withering, green removing, rolling and twisting, and sun-drying, and marked corresponding numbers according to the samples names before grinding. All samples were stored at -4℃ until further analysis."

Author Response

Thanks very much for taking the time to review this manuscript. I really appreciate all you comments and suggestions! Please see the attachment.
